# Antimicrobial Pharmacokinetics and Pharmacodynamics in Critical Care: Adjusting the Dose in Extracorporeal Circulation and to Prevent the Genesis of Multiresistant Bacteria

**DOI:** 10.3390/antibiotics12030475

**Published:** 2023-02-27

**Authors:** Jesus Ruiz-Ramos, Laura Gras-Martín, Paula Ramírez

**Affiliations:** 1Pharmacy Department, Hospital Santa Creu i Sant Pau, 08025 Barcelona, Spain; 2Intensive Care Unit, Hospital Universitario y Politécnico La Fe, 46026 Valencia, Spain

**Keywords:** pharmacokinetics, antibiotic resistance, antibiotic activity, intensive care unit, multidrug-resistant pathogens

## Abstract

Critically ill patients suffering from severe infections are prone to pathophysiological pharmacokinetic changes that are frequently associated with inadequate antibiotic serum concentrations. Minimum inhibitory concentrations (MICs) of the causative pathogens tend to be higher in intensive care units. Both pharmacokinetic changes and high antibiotic resistance likely jeopardize the efficacy of treatment. The use of extracorporeal circulation devices to support hemodynamic, respiratory, or renal failure enables pharmacokinetic changes and makes it even more difficult to achieve an adequate antibiotic dose. Besides a clinical response, antibiotic pharmacokinetic optimization is important to reduce the selection of strains resistant to common antibiotics. In this review, we summarize the present knowledge regarding pharmacokinetic changes in critically ill patients and we discuss the effects of extra-corporeal devices on antibiotic treatment together with potential solutions.

## 1. Introduction

Serious infections are both the cause and consequence of critical conditions in intensive care units (ICUs). Once an infection is diagnosed, the appropriate (by antibiogram) and adequate (by pharmacokinetic and pharmacodynamic parameters (Pk/Pd)) administration of antibiotics is the primary intervention that the attending physician can use to improve patient prognosis [1]. In critically ill patients, frequent pathophysiological changes can alter the pharmacokinetic parameters of drugs and, therefore, change the bioavailable concentrations in the blood and at the site of infection [2]. Depending on the antibiotic pharmacodynamic objective and the minimum inhibitory concentration (MIC) of the causative microorganism, this can result in therapeutic failure [3]. Furthermore, the increase in the frequency of multi-drug-resistant microorganisms with high MICs has intensified the magnitude of this situation.

The use of extracorporeal circuits to support or replace dysfunctional organs may further change Pk properties, making antibiotic treatment failure more likely.

In this article, we review the pathophysiological changes that affect the pharmacokinetics (Pk) of antibiotics in critically ill patients, the effect that the use of extracorporeal circulation can have on drugs, and how to achieve Pk/Pd objectives.

## 2. Pharmacokinetic Variations in Critical Patients

The most common Pk changes in critically ill patients are an increased antibiotic-free fraction resulting from hypoalbuminemia, an increased volume of distribution (Vd), and liver and kidney dysfunction, which either alter renal clearance or cause acute renal failure and antibiotic accumulation [2]. Based on the hydrophilic or lipophilic nature of a drug, these changes will be more or less relevant. In general, hydrophilic drugs will be more affected [3,4,5]. Secondly, a Pk/Pd objective can modulate posology strategies to overcome critical pathophysiological disturbances [6,7]. Importantly, a loading dose of hydrophilic antibiotics is highly recommended [8].

Currently, adjusting dose regimens through the therapeutic drug monitoring (TDM) of plasma concentrations has become increasingly recommended, being considered an essential element in the management of septic patients [1,9,10,11]. Table 1 lists the major antibiotic Pk/Pd properties and their common alterations in critically ill patients.

## 3. Renal Replacement Therapy

Critically ill patients may require renal replacement therapy (RRT) support and a therapeutic adjustment is essential to avoid treatment failure, toxicity, or antibiotic drug resistance. In recent years, many patients undergoing RRT did not achieve therapeutic goals at standard doses [40]. Among the RRT factors that can significantly affect clearance (CL) of the antibiotic, the use of different modalities of solute removal and prescribed intensity (blood flow rate and effluent flow rate), filter material, filter surface, and duration of RRT should be considered [41].

Intermittent hemodialysis results in a temporary reduction in plasma levels of large Vd drugs, which is followed by a post-treatment “rebound.” Meanwhile, in continuous RRT (CRRT), clearance and redistribution equilibrium is commonly established [42].

The three main modalities of CRRT include hemofiltration (CVVH), hemodialysis (CVVHD), and hemodiafiltration (CVVHDF), based on convection, diffusion, or both, respectively. Clearance efficiency is higher in CVVHDF, followed by CVVHD, and finally, CVVH [41,43]. Glomerular filtration rates have been described for each modality, although they are very general and depend on other factors. These are 15–25 mL/min for CVVH and CVVHD, 30–40 mL/min for CVVHDF, and highly variable (between 10 to 50 mL/min) for sustained low-efficiency dialysis (SLED) [16], which results in different rates of antibiotic elimination.

The membrane type (polysulfone, polymethylmethacrylate, and polyacrylonitrile membranes) and membrane surface area can also impact antibiotic CL. The membranes used for intermittent hemodialysis usually have small pores and do not allow the removal of molecules larger than 500 daltons (Da). During CRRT, large pores increase the size of the dialyzed molecules (>1000 Da). In general, the effect of adhesion to the membrane is negligible for most antibiotics. However, some studies describe a decline in serum concentrations for piperacillin, amikacin, or levofloxacin [44,45].

Another CRRT factor that plays an important role in the CL is the prescribed flow [43,46]. Various studies have reported a significant relationship between effluent flow and CL for several antibiotics in CRRT. Some of these include meropenem, piperacillin-tazobactam, ceftolozano-tazobactam, meropenem-vaborbactam, vancomycin, and dalbavancin [41,46,47]. However, the results of these studies have not been consistent [48,49]. Filter efficiency is lost over days until circuit components are changed and, therefore, antibiotic CL efficacy could change over days [46].

The primary patient-related factors that can increase total CL in patients with CRRT include hypoalbuminemia, increased hepatic and biliary metabolism, and residual diuresis. One study also found a relationship between hematocrit and the efficiency of the drug transfer through the filter [50]. Patients who undergo CRRT may have some degree of preserved residual renal function [44,49,50,51,52]. The evaluation of renal functions using formulas, such as Cockcroft–Gault, are not reliable in this instance, whereas measuring creatinine clearance in urine (short time collection, 3–4 h) and Cystatin C are the most reliable methods for assessing glomerular filtration measurement [44,46,53].

On the other hand, it is essential to take into account the Pk characteristics of the drugs to assess the measure through which they will be cleared by RRT. In general, molecules with high molecular weight, a high binding rate to plasma proteins (>80%), lipophilic antibiotics, high Vd (>1–2 L/kg), and nonrenal clearance antibiotics, will have low or no elimination through the hemofilter [46]. Nevertheless, some circumstances can increase the free drug rate, binding to plasma proteins, hyperbilirubinemia, uremic toxins, and an alteration of blood pH. As a result, RRT clearance will increase in the same proportion [42]. For diffusive techniques, the dialysate ratio (HDCVV) is more strictly dependent on molecular weight (MW) and elimination is generally decreased for agents with a MW > 1000 to 1500 Da, although most antibiotics are <1000 Da except for glycoproteins, lipo-proteins, and colistin [41].

Finally, the interaction between the drug and the electrical charges of the dialysis membrane is described by the Gibbs–Donnan effect, in which anionic proteins present in the blood, such as albumin, tend to retain cationic molecules (aminoglycosides and levofloxacin), while facilitating the passage of anionic drugs (some cephalosporins, such as ceftazidime and cefotaxime) [44]. However, the clinical significance of this interaction does not appear to be relevant.

The sieving coefficient (SC), which is a specific parameter for each antibiotic for a given membrane, has been used to determine the CL in the RRT. The value of the SC goes from 1, which indicates a high clearance in RRT, to 0, which indicates no RRT clearance [54]. Different equations have been used to estimate the antibiotic CL based on the SC and the modality of renal replacement used [42]. However, SC calculation can be extremely complex, especially for HDCVV and HDFCVV, in which factors, such as the saturation coefficient, the efflux rate, and hypoalbuminemia, can modulate the SC [41].

In clinical practice, there is a wide heterogeneity in the CRRT dose, blood and dialysate flow, types of filters and dialysis, and the surface membranes selected. Therefore, antibiotic dose optimization is highly challenging. Some recommendations have been published, but not all of them have been validated in clinical practice, nor do they consider all of the factors that may affect CRRT [46]. In general, loading doses depend only on the Vd and do not require any adjustment, even in patients with severe renal insufficiency [42]. The information regarding elimination by CRRT and the doses described for each antibiotic are listed in Table 1.

## 4. Extracorporeal Membrane Oxygenation

There is a high risk of infection during extracorporeal membrane oxygenation (ECMO), which is associated with high mortality (up to 50% in some publications) [55]. Therefore, antibiotic therapy is commonly prescribed in ECMO patients, but data detailing the appropriate dosage are very limited and of low quality. Most studies have been carried out in the pediatric setting or more specifically, the neonatal population, and extrapolation of these results to adults can be erroneous [56].

ECMO support can influence the Pk and, therefore, the achievement of the Pk/Pd target. The main changes include variability in the elimination of the drug (generally described as a decrease) and an increase in the Vd resulting from hemodilution (especially in hydrophilic drugs) or sequestration of the drug in the circuit (especially for lipophiles and antibiotics with high binding to plasma proteins) [57]. The intensity of the effect of ad-herence to the circuit may also be influenced by the MW and the degree of ionization of the drug, although this is less studied. The impact of the ECMO on the Vd is difficult to estimate because the inflammatory process resulting from ECMO treatment or caused by sepsis and the patient’s illness may also contribute to its variability [57,58,59]. The type of oxygenator and pump, cannula materials, and primer solution composition can also affect drug removal. The silicone membrane may result in greater drug adherence compared with other oxygenators (hollow fiber, pink membrane oxygenators) as well as the loss of labile drugs at 37 °C [56]. A membrane exchange could therefore indicate the need for a new loading dose [56].

Renal and hepatic CL appears to decrease in ECMO patients because of reduced organ perfusion, although the initial hyperdynamic state of the critically ill patient with intense fluid therapy and vasoactive agents could increase the initial CL [60].

In recent years, there have been efforts to determine the variations in some drugs in patients with ECMO; however, many studies have not been able to optimize treatment posology for these patients. Furthermore, many ECMO patients are also treated with RRT, which suggests the importance of TDM [2,56,59]. However, studies show varying and unexpected results. No differences in pharmacokinetic parameters have been observed in clinical practice for vancomycin, piperacillin-tazobactam, meropenem, azithromycin, quinolones, amikacin, and tigecycline [56]. In contrast, there are considerable changes in the Pk of imipenem [61]. The information on changes by ECMO and the doses described for each antibiotic are listed in Table 1.

## 5. Antibiotic Therapeutic Drug Monitoring

In recent decades, antibiotic therapeutic drug monitoring (TDM) has evolved from toxicity prevention of drugs with a narrow therapeutic index to an essential tool to improve the response to antibacterial treatment and prevent the emergence of resistance. There is now a strong rationale to individualize antibiotic dosing in critically ill patients with the aid of TDM [62,63], being positioned as a valuable tool to improve the clinical results of patients with a severe infection.

Several studies have shown that the application of TDM leads to improvements in Pk/Pd achievement, leading to improved clinical outcomes for patients [64,65]. However, the impact of the TDM of new antibiotics on the clinical evolution of patients and the generation of resistance needs to be evaluated in the coming years, in order to find the best Pk/Pd values to apply in the different types of infection.

## 6. Pk/Pd to Suppress the Emergence of Bacterial Resistance

Optimization of Pk/Pd parameters to minimize antibiotic resistance has not received sufficient attention in clinical practice. Pharmacokinetic goals are generally focused on maximizing clinical and microbiological outcomes, without considering resistance suppression. Nevertheless, given the progressive increase in resistance observed at the usual doses for most antibiotics, clinical data are needed to define thresholds that can minimize the emergence of resistance without compromising patient safety.

Acquired antibiotic resistance mechanisms can be divided into four main categories: modifying the antibiotic target (PBPs alterations), limiting uptake (decreased numbers of porins), antibiotic inactivation (β-lactamases) and active drug efflux pumps. Gram-negative bacteria can use all four main mechanisms, whereas gram-positive bacteria less commonly use limiting the uptake of a drug, and certain types of drug efflux mechanisms are not usable. Clinical and in vitro studies have shown that the low antibiotic dose and treatment duration can influence the selection of antibiotic-resistant mutants [66]. However, the impact of an antibiotic dosage or duration in specific mechanisms of resistance expression is uncertain.

There is poor knowledge of the optimal dosing strategies to treat bacterial infections while simultaneously preventing the selection and emergence of resistance. It is known that a subpopulation of resistant bacteria often exists and can be selected at certain drug concentrations, leading to a regrowth during treatment. Low antibiotic concentrations can select for low-level resistance, which could have a major effect on the emergence of high-level antibiotic resistance [67]. On the other hand, during antibiotic treatment, a selection of resistance may take place at several sites. Therefore, in the evaluation of optimal drug concentrations, it is important to focus not only on the infective pathogens and the infectious sites, but also on the commensal flora (e.g., intestinal tract), in which much higher numbers of bacteria exist and which perhaps is even more important in the selection of resistance.

The Pk/Pd index relates antibiotic exposure to the antibiotic susceptibility of an infecting pathogen, in which susceptibility is described as the MIC, thus providing a dosing target. However, in vitro studies simulating current antibiotic dosing highlight the limited ability to suppress the emergence of antibiotic-resistant bacteria [68]. Because the MIC is a measure of susceptibility for most of the bacterial population at a standardized inoculum, alternative measures of susceptibility are needed to determine the risk of devolving antibiotic resistance to provide new Pk/Pd targets for suppressing resistance emergence.

The mutant prevention concentration (MPC), which is defined as the lowest concentration that blocks the emergence of first-step resistant mutants in a large susceptible population [69], has been proposed as a cutoff point to select an antibiotic concentration that prevents antibiotic resistance. Antibiotic concentrations ranging between the MIC and the MPC are known as the mutant selection window (MSW) and promote the growth of resistant bacterial pathogens [70]. Therefore, the antibiotic exposure required to suppress the emergence of resistance should be maintained above the MSW [71]. However, no standardized definitions exist to determine the antibiotic exposures that should be targeted to suppress the emergence of antibiotic resistance. On the other hand, the MPC is based on the concept that antibiotic resistance is a function of sequential mutations; therefore, preventing the first mutation will effectively prevent subsequent mutations. The MPC has mostly been studied for antibiotics for which resistance primarily develops by stepwise chromosomal point mutations, especially the fluoroquinolones. The application of MPC determination to other drugs has raised questions regarding the relevance of mutational events for such drugs as the β-lactams and the aminoglycosides and whether MPC measurements can be performed for drugs with other resistance mechanisms (efflux pumps and β-lactamases) [72,73]. Mutations that can lead to resistance may arise in many different combinations, and the correlation between the MIC and the MPC is probably dependent on where the mutation is likely to arise. Table 2 summarizes the main Pk/Pd objectives proposed to prevent the development of resistance as well as the potential adverse effects associated with high doses of the various antibiotic groups.

### 6.1. β-Lactams

Preclinical studies have indicated that exposures of Cmin/MIC ≥ 6–8 suppress the emergence of resistance of highly susceptible isolates [74,75]. Other studies have demonstrated that resistance emergence may occur with an exposure of Cmin/MIC < 4 against high bacterial densities [76]. In contrast, the exposure needed for optimal clinical cure varies between fT > MIC > 40% and a Cmin/MIC ≥ 4, depending upon the infection type and severity [77,78]. Thus, the antibiotic dose required to suppress the emergence of β-lactam antibiotic resistance in most patients is higher compared with that required for a preclinical effect [79].

### 6.2. Fluoroquinolones

Fluoroquinolone use is strongly associated with the development of resistance. Some in vitro studies have suggested that the index associated with the suppression of resistance is an AUC/MIC ratio > 200 [80]. However, consideration should be given to the potential risks of high-dose therapy, which may include cardiac dysrhythmias and confusion. Combined therapy may be the best strategy to reduce the emergence of resistance, although contradictory results have been published.

### 6.3. Aminoglycosides

The Pk/Pd target for preventing the emergence of resistance depends on the specific molecule. With a once-daily administration, a lower dose is necessary to suppress re-sistance compared with a two-dose a day schedule (Cmax/MIC ratio 13 vs. Cmax/MIC ratio 20) [81]. In vitro models have indicated that when longer concentrations were maintained above the MPC (time that the concentration was inside the mutant selection window), there was a lower enrichment of resistant subpopulations [82].

### 6.4. Fosfomycin

The Pk/Pd parameter that is associated with a lower risk of resistance is the AUC/MIC ratio. In an in vitro model, an AUC 0–24/MIC ratio > 3136 was adequate for the suppression of resistance development in E. coli with a MIC of 1 mg/L. and a daily dose of 24 g was needed to achieve the Pk/Pd target [83]. However, given the quick development of resistance observed after the use of this drug, fosfomycin should only be administered in combination and at high daily doses to prevent resistance.

### 6.5. Colistin

The Pk/Pd parameter that best describes the success of therapy and probability of suppression of the emergence of resistance with colistin is the AUC/MIC ratio. In a clinical setting, when a 9 M IU loading dose was given, followed by 4.5 MIU/8 h, no resistance emerged in 127 ICU patients [84]. However, in vitro models have indicated that no dose of colistin or polymyxin B could suppress the emergence of resistance [85]. One study in the murine tight and lung infection model revealed that *A. baumannii* developed resistance even at a colistin exposure > 10 mg/L, which is a much higher concentration compared with the usual clinical dose [86]. Because of its high probability of inducing resistance, colistin should never be administered alone, even at high doses.

### 6.6. Linezolid

Few studies have described the linezolid Pk/Pd ratios required to prevent the emergence of resistance. An AUC/MIC > 124 was shown to suppress resistance against clinical MRSA isolates. Another study indicated that the dose required to achieve a linezolid steady-state concentration approximately equivalent to the pathogen MIC may promote the emergence of resistance [87]. These data indicate that low linezolid exposure in isolates without baseline resistance mechanisms may result in the emergence of resistance.

### 6.7. Daptomycin

A total AUC/MIC > 200 has been associated with an avoidance of *S. aureus* dap-tomycin resistance emergence in a dynamic one-compartment in vitro infection model [88]. A dose of 6 mg/kg/day resulted in resistance emergence in an in vitro model and a clinical case report [89], possibly because of enhanced daptomycin clearance, which occurs in critically ill patients [90].

### 6.8. Glycopeptides

Various studies have identified vancomycin AUC/MIC > 200 as sufficient to suppress resistance emergence in an *S. aureus* in vitro infection model [91], suggesting that the current targeting dose of AUC/MIC > 400, which is associated with improved clinical outcomes, is sufficient to avoid resistance emergence. However, Lenhard et al. [92] demonstrated that an AUC/MIC > 1800 was required to suppress vancomycin resistance against two MRSA isolates. The initial inoculum differed between the studies, suggesting that the exposure required to suppress resistance emergence depends, not only on the specific isolate, but additionally on the bacterial inoculum [93].

## 7. Machine Learning, Big Data, and Artificial Intelligence

In recent years, artificial intelligence and machine learning techniques have emerged as new tools to predict the risk, and even to prevent, colonization and infection by MDR bacteria as well as the development of resistance associated with the use of antibiotics [94,95]. These techniques have additionally been used to create support tools capable of learning classification rules to identify inappropriate prescriptions and recommend dose adjustments. The enormous potential of these tools and the continued exploration of their usefulness in daily practice will occur over the coming years. The opportunity of using machine learning predictions for a drug Pk as an input for a Pk/Pd model may accelerate their clinical development.

There are some examples in the literature in which machine learning has been used to predict Pk data [96,97]. These studies showed that integrating an artificial network with the usual Pk population algorithms can predict the best drug concentration over time. Smith et al. [98] found that a data-driven, model-informed process could determine the optimal treatment strategy, including dose, infusion rate, number of daily doses, and loading dose. Using machine learning analysis, Alsaher MH et al. [99] identified Pk/Pd parameters capable of predicting clinical cure. According to this model, early and cumulative target attainment could have a significant impact on pneumonia outcomes. Using a machine learning approach to integrate clinical data into a predictive model for initial vancomycin dosing has also been evaluated [100]. A predictive model with a target attainment rate comparable to that achieved by experts in Pk was obtained. These strategies may be used to develop a predictive model that could determine initial antibiotic dosing, particularly in settings in which dose-planning consultations are unavailable.

### 7.1. Big Data Analysis

Big data analysis for Pk/Pd optimization has also received significant attention recently [101,102]. Current Pk/Pd models for critically ill patients are usually based on a limited number of patients, which leads to the exclusion of clinical parameters because of its limited contribution to the final model. In this context, models based on big data, including a huge amount of data from electronic medical record systems, are an interesting new approach to Pk research. These programs should be able to calibrate models based on the available data and interventions and become more robust over time, being the first step in applying artificial intelligence to closed loop systems. This model will help to identify an optimal dose based on a patient’s clinical situation and consider the probability of the different pathogens responsible for infection as well as identify risk factors for altered Pk/Pd parameters to better predict those patients at risk of underdosing or overdosing.

### 7.2. Bedside Antibiotic Monitoring Systems

Despite the high variability in antibiotic concentrations and the risk of underdosing in critically ill patients, most patients continue to receive standard dosing regardless of the underlying disease or existing comorbidities. Although therapeutic antibiotic monitoring has grown significantly in many hospitals, it has produced mixed results and has major downsides, including the time consumed until dose optimization following the initiation of antibiotic therapy.

The development of real time closed-loop decision support systems at the bedside may be a major step forward [103], even with advice before the first dose is prescribed. This may help the next generation of critical care physicians to prescribe the optimal dose to the right patient at the right time in any situation. First experiences are being developed, which will offer promising results [101].

## 8. Conclusions

The complexity of the critically ill patient affects the Pk/Pd of antibiotics and can lead to an erroneous dosing and a poor prognosis. Currently, specialists in the treatment of infection in critically ill patients must be aware of all aspects concerning the patient, the infectious foci, the microorganism, and the drug involved in the episode, to properly modify the antibiotic dosage regimen. Special attention should be paid to the study of the notorious effects that extracorporeal circulation systems have on the Pk of most antibiotics. Because of the consequences of the progressive increase in the rate of multidrug-resistant strains, pharmacokinetic optimization must be applied to avoid the emergence of resistant strains. In the near future, artificial intelligence and ML techniques may enable the optimization of antibiotic treatment and improve overall patient prognosis.

## Figures and Tables

**Table 1 antibiotics-12-00475-t001:** Characteristics of antibiotics and variations in critically ill patients, CRRT and/or ECMO.

Drug	Renal Elimination	Pk/Pd Target	Dosage Regimen Proposed in cRRT		Dosage Regimen Proposed in ECMO
Time dependent antibiotics					
β-lactams		fT > 100% MIC or fT > 100% 4 × MIC	fT > 1 × MIC (2–4 mg/L)	fT > 4 × MIC or high MIC (4–16 mg/L)	
Ceftriaxone	30–60%		2 g/24 h	2 g/24 h [12]	Standard dose
Meropenem	70–80%		1 g/12 h, 500 mg/8 h 3 h PI or 500 mg/6 h. 1 g/8 h high flow rate [12,13]	1 g/6 h 30 min or 500 mg/6 h 3 h PI [12,13]	Standard dose
Imipenem	70%		500 mg/6 h or 1 g/6 h high flow rate [12]	1 g/6 h [12]	Standard dose
Piperacillin/Tazob.	70–75%		4 g/6 h or 12 g CI [14]	4 g/6 h or 12 g CI [14]	Standard dose
Ceftazidime	85%		1 g/12 h [13]	1250 mg/8 h or 1500 mg/8 h high flow rate [13]	Standard dose
Cefepime	85%		2 g/24 h low flow rate or 2 g/12 h high flow rate	1 g/8 h or 2 g/12 h low flow rate; 2 g/8 h 3 h PI or 1 g/6 h or 4 g/24 h CI high flow rate [12,13]	1 g/8 h or 1 g/12 h [15]
Ceftolozane/Tazob.	85%		0.5/0.25 g/8 h or 1/0.5 g CI low flow rate; 1/0.5 g/8 h or 1.5/0.75 g CI high flow rate [12]	1/0.5 g/8 h or 1.5/0.75 g CI; 2/1 g/8 h IC high flow rate [12]	Standard dose
Ceftazidime/Avib.	85%		1/0.25 g/12 h [13]	1/0.25 g/8 h or 2/0.5 g/8 h [12,13]	Standard dose
Ceftaroline	85–90%		400 mg/12 h [16]	400 mg/12 h [16]	No data
Cefiderocol	90%		1.5 g/12 h [17]	1.5 g/12 h or 2 g/8 h high flow rate [17]	No data
Clindamycin	10–15%	AUC/MIC or fT > 100% MIC	600–900 mg/6–8 h		No data
Fosfomycin	85%	fT > 100% MIC or fT > 100% 4 × MIC.	8 g/12 h or 4 g/6 h or 16 g CI [18]		No data
Concentration-dependent antibiotics					
Aminoglycosides	>95%	Cmax/MIC > 8–12	Amikacin 25 mg/kg/48 h; Gentamicin and Tobramicin 7–8 mg/kg/48 h		Standard dose [19]
			If high effluent (≥35 mL/kg/h) assess c/24 h [20]		
Concentration-dependent with time-dependent antibiotics					
Azithromycin	10–15%	AUC/MIC > 5	500 mg/24 h [21]		Standard dose [22]
Colistin	60–70%	AUC/MIC > 50–65	3 MUI/8 h or 4.5 MUI/12 h [12,23]		Standard dose [24]
Cotrimoxazole	80% Trimethoprim	Cmax/MIC and AUC/MIC	5 mg Trimethoprim/kg/12 h [25]		Standard dose [26]
Glycopeptides					
Teicoplanin	65%	AUC/MIC ≥ 750 (CmIn 10–20 mg/L) or AUC/MIC ≥ 1500 (≥20–30 mg/L)	10 mg/kg/12 h 4 doses and then if:		High standard dose [27]
			CVVH: 10 mg/kg/48 h		
			CVVHD: 8 mg/kg/24 h		
			CVVHDF: 6 mg/kg/24 h		
			If high effluent rate (30–35 mL/kg/h), increase dose 30%. [28]		
Vancomycin	80%	AUC/MIC > 400	15–22 mg/kg/24 h if low effluent rates		High standard doses (higher doses than those recommended in renal dysfunction) [29]
			15 mg/kg/12 h high effluent rate [30]		
			If residual diuresis > 0.5 mL/kg/h and effluent rate 0.6–3 L/h: 12.2–23.1 mg/kg/12 h. [31]		
			500 mg/12 h CVVH and 500 mg/8 h CVVHD [12]		
Lipoglycopeptides					
Dalbavancin	20–35%	AUC/MIC > 111	No data		No data
Daptomycin	60%	AUC/MIC ≥ 666	6–8 mg/kg/24 h [12,32]		10 mg/kg/24 h [33,34]
Oxazolidinones					
Linezolid	30–50%	85%T > MIC y AUC/MIC > 80–120	600 mg/8–12 h (900 mg/8 h used by some authors) [35]		600 mg/8–12 h [36]
Tedizolid	15–20%	AUC/MIC > 3	No data		No data
Quinolones					
Ciprofloxacin	65%	AUC/MIC > 125 or Cmax/MIC > 10 Gram-negative; AUC/MIC > 25–30 Gram-positive	400 mg/12 h or 200–400 mg/8 h [12,37]		400 mg/24 h [38]
Levofloxacin	65%		250 mg/24 h [12]		Standard dose
Moxifloxacin	20–25%		No data		No data
Tigecycline	10–15%	AUC/MIC:	50 mg/12 h; Pneumonia: 100 mg/12 h	100 mg/12 h	Standard dose [39]
		Intrabdominal > 6.96			
		Pneumonia > 10.1			
		Soft-tissue > 17.9			

CI: Continuous infusion; CVVH: Continuous Venovenous Hemofiltration; CVVHD: Continuous Venovenous Hemodialysis; CVVHDF: Continuous Venovenous Hemodiafiltration; PI: Prolonged infusion.

**Table 2 antibiotics-12-00475-t002:** Main pharmacokinetic/pharmacodynamic (Pk/Pd) index associated with resistance prevention and sides effects related with high antibiotic concentrations.

Antibiotic Group	Pk/Pd Index	Pk/Pd Suggested to Prevent Resistance	Clinical Pk/Pd Threshold for Toxicity	Main Side Effect Expected at High Dose
β-lactams				
Carbapenems	%T > MIC	Cmin/MIC ≥ 6–8	Cmin > 44.5 mg/L	Neurotoxicity
Cephalosporins	%T > MIC	Cmin/MIC ≥ 6–8	Cmin > 20 mg/L	Neurotoxicity
Penicillins	%T > MIC	Cmin/MIC ≥ 6–8	Cmin > 361 mg/L	Neurotoxicity
Aminoglycosides	Cmax/MIC	Cmax/MIC ≥ 13	Amikacin: Cmin > 2.5 mg/L Gentamicin, Tobramicin: Cmin > 0.5 mg/L	Nephrotoxicity
Daptomycin	AUC/MIC	AUC/MIC ≥ 200	Cmin ≥ 24.3 mg/L	Myopathy
Fluoroquinolones	AUC/MIC	AUCMIC ≥ 200	Unclear	QT prolongation, dysrhythmias, Neurotoxicity, gastrointestinal disorders
Glycopeptides	AUC/MIC	AUC/MIC > 400–1800	Cmin > 20 mg/L	Nephrotoxicity
Linezolid	AUC/MIC	AUC > 124 mg/L/h	Cmin > 7 mg/L	Hematological toxicity
Polymyxins *	AUC/MIC	Cmin ≥ 10 mg/L	Cmin > 2.4 mg/L	Nephrotoxicity
Fosmomycin *	AUC/MIC	AUC/MIC > 3136	Unclear	Hypernatremia, gastrointestinal disorders

AUC: Area under the curve; Cmax: Maximum concentration (Post-dose); Cmin: Minimum concentration (Pre-dose); MIC: Minimum inhibitory concentration; %T > MIC: % Time above MIC; * Combined therapy is needed.

## Data Availability

Not applicable.

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
