# Peer review of "Antimicrobial Pharmacokinetics and Pharmacodynamics in Critical Care: Adjusting the Dose in Extracorporeal Circulation and to Prevent the Genesis of Multiresistant Bacteria"

_antibiotics, 2023, doi:10.3390/antibiotics12030475_

Round 1
Reviewer 1 Report
The review submitted by Ruiz-Ramos et al discusses changes in pharmacokinetics of important antibiotics in critically care patients. The review discusses recent data on the use of antibiotics of different classes in patients with several critical health issues. Information on the changes in PK of antibiotics is provided and some suggestions on addressing the issue are made.
I have following comments to the text.
1) Abstract: I'd recommend either to remove or explain all abbreviations used in abstract, such as Pk/Pd, pK, ICU.
Also, all abbreviations should be made uniform as in the current version Pk, pK and PK are used for pharmacokinetics.
2) The language of the manuscript needs checking and correction. This version of the manuscript contains many errors.
3) Table 1 is hardly readable. I recommend its redesign and use the landscape page layout for presentation.
The abbreviations should be explained. What is cRRT? ECMO?, fT? MIC?
The objective of providing the data in Table 1 is not clear. The text postulates it to contain PK alterations of antibiotics in critically ill patients, but in this case I suggest providing the data on a dosage regime(s) used normally. Are all columns necessary which are given? The column "ECMO alterations" consist only of "not relevant" or "unclear" entries.
4) The objective of providing the data in Table 1 is not clear. The text postulates that Table 1 contains PK alterations of antibiotics in critically ill patients, but in this case I suggest providing the data on a dosage regime(s) used normally. Are all columns necessary which are given?
5) Page 5: The subheading "Concentration-dependent with time-dependence antibiotics" is used twice.
6) Page 5: hours or h should be used instead of "horas".
7) Page 6: "dose" but not "doce".
8) Page 11: The term "Machine learning" should be used here and throughout the text.
9) Conclusions: "must be aware" but not "we aware".
In general, I recommend second revision of the text and thorough revision of presenting information in Table 1.
Author Response
The review submitted by Ruiz-Ramos et al discusses changes in pharmacokinetics of important antibiotics in critically care patients. The review discusses recent data on the use of antibiotics of different classes in patients with several critical health issues. Information on the changes in PK of antibiotics is provided and some suggestions on addressing the issue are made.
I have following comments to the text.
Abstract: I'd recommend either to remove or explain all abbreviations used in abstract, such as Pk/Pd, pK, ICU. Also, all abbreviations should be made uniform as in the current version Pk, pK and PK are used for pharmacokinetics.
R: According to this suggestion, abbreviations have been removed form the abstract. We have also made uniform the abbreviation Pharmacokinetic as Pk
The language of the manuscript needs checking and correction. This version of the manuscript contains many errors.
R: A native-speaker English has reviewed the manuscript. Changes have been made in order to correct grammatical errors, typos and achieve a better understanding.
Table 1 is hardly readable. I recommend its redesign and use the landscape page layout for presentation.The abbreviations should be explained. What is cRRT? ECMO?, fT? MIC?The objective of providing the data in Table 1 is not clear. The text postulates it to contain PK alterations of antibiotics in critically ill patients, but in this case I suggest providing the data on a dosage regime(s) used normally. Are all columns necessary which are given? The column "ECMO alterations" consist only of "not relevant" or "unclear" entries.The objective of providing the data in Table 1 is not clear. The text postulates that Table 1 contains PK alterations of antibiotics in critically ill patients, but in this case I suggest providing the data on a dosage regime(s) used normally. Are all columns necessary which are given?
R: We agree with this commentary. Table 1 has been redesigned in order to a better understanding. Abbreviations have been included in the footnote.
Page 5: The subheading "Concentration-dependent with time-dependence antibiotics" is used twice.
R: Change has been made in the manuscript to fix this typo
Page 5: hours or h should be used instead of "horas".
R: Change has been made in the manuscript to fix this typo
Page 6: "dose" but not "doce".
R: Change has been made in the manuscript to fix this typo
Page 11: The term "Machine learning" should be used here and throughout the text.
R: According to this suggestion, the abbreviation ML has been changed to “Machine learning”
Conclusions: "must be aware" but not "we aware".
R: Change has been made in the manuscript to fix this typo

Reviewer 2 Report
This paper described the review about pharmacokinetics and pharmacodynamics (PK/PD) of antibiotics in critically ill patients, that frequent pathophysiological changes could modify the PK parameters of the antibiotics. Overall, this paper is organized well, however, there are some comments.
1) It is curious that there is no mention of sepsis or septic shock in this article. The authors have to refer to such as “Surviving Sepsis Campaign: International Guidelines for Management of Sepsis and Septic Shock 2021”, and make mention of PK/PD of antibiotics in patients with sepsis or septic shock.
2) Cystatin c have also to be mentioned on the evaluation of renal function.
3) The authors have to cite references in the description of the introductory section.
4) Table 1 is difficult to read and understand. It must be revised.
Author Response
This paper described the review about pharmacokinetics and pharmacodynamics (PK/PD) of antibiotics in critically ill patients, that frequent pathophysiological changes could modify the PK parameters of the antibiotics. Overall, this paper is organized well, however, there are some comments.
It is curious that there is no mention of sepsis or septic shock in this article. The authors have to refer to such as “Surviving Sepsis Campaign: International Guidelines for Management of Sepsis and Septic Shock 2021”, and make mention of PK/PD of antibiotics in patients with sepsis or septic shock.
R: We agree with the reviewer that this is an important reference to be included. The following sentence has been including in the manuscript, citing the Survival sepsis Campaign guideline
“Currently, adjusting dose regimens through the therapeutic drug monitoring (TDM) of plasma concentrations has become increasingly recommended, being considered an essential element in the management of septic patients [8]”
Cystatin c have also to be mentioned on the evaluation of renal function.
We appreciate this suggestion. The following sentence and reference have been included in the manuscript.
“The evaluation of renal function using formulas, such as Cockcroft–Gault, are not reliable in this instance, whereas measuring creatinine clearance in urine (short time collection, 3–4 h) and Cystatin C are the most reliable methods for assessing glomerular filtration measurement [18,20,27].”
Ref: Maheshwari KU, Santhi S, Malar RJ. Cystatin C: An alternative dialysis adequacy marker in high flux hemodialysis. Indian J Nephrol. 2015 May-Jun;25(3):143-5. doi: 10.4103/0971-4065.139489.
The authors have to cite references in the description of the introductory section.
R: References have been included in the introductory section
Table 1 is difficult to read and understand. It must be revised.
R: According to this suggestion, Table 1 has been reviewed and readapted for a better understanding.

Reviewer 3 Report
The paper by Ruiz-Ramos et al. reviews the literature regarding PK/PD issues in relation to the critically ill patient undergoing the various types of renalreplacemnet therapy including ECMO including a view on the dosing in regard to resistance development. There are recent reviews in this area, but given the complexity of the subject it is worthwhile to update our knowledge focusing on new studies.
Comments:
1. The manuscript needs serious English revision. It is too laborious to mention all mistakes, but here are some: Abstract Line 12: ..antimicrobials are one of the most frequently used drugs..; line 14: plasmatic; Introduction, line 46: ..improve the achievement Pk/Pd...; Line 51: The most commons Ph changes.. etc.etc.
2. The authors focus on antibiotics and bacteria only, fungi are not mentioned,, why they should stick to this subject and use this term alone, not antimicrobials, and bacteria in stead of microorganisms.
3. Pharmacokinetics is abreviated to Pk (use PK), then this should be used throughout the text and not introduce the term "pharmacokinetics" several times in the text.
4. Table 1 is virtually unreadable. The table should be written vertically to allow space for all columns. Alle abbreviations must be explained either in the legend or in foot notes. The term "f" for "free" (protein non-bound) is only used for T>MIC in the start, and not explained. If this is used, it should also be used for Cmax and AUC in the rest of the table - or do not use it but expalin in the legend that all parameters are "free" concentrations.. fT>100%MIC should be 100% fT>MIC. Ceftazidime is once spelled ceftazidima, and Taz should be written out as tazobactam, since other inhibitors are written out. Explain why T/MIC or T/ 4 x MIC is used ?
5. The manuscript part on prevention of resistance is too simple. The types of resistance for the different drug classes should be explained. It is not clear to me, which type of resistance occurs for beta-lactams (efflux?), while this is more clear for fluoroquinolones. The same goes for resistance towards vancomycin in staphylococci, which is debatable. I thought resistance seldom occurred with gentamicin, while it was common for streptomycin. Also, the importance of high bacterial burden is an issue for the effect of most antibiotics, not just for selection of resistance, but for sheer effect of the antibiotics (persisters and others). In this context it might also be mentioned, that the amount of antibiotics used is also a factor related to selection of resistance, and improving PK/PD should in theory improve the effect and at the same time reduce the duration of therapy, which in itself reduces the exposure on the normal flora.
6. Table 2: What is the meaning of the "n" after the first MIC in the equation ?
7. There is very little text on the issue of TDM - only about bedside monitoring. With all the importance of the the PK/PD target it would be logical to discuss the approriateness on more TDM for beta-lactams, while it is standard for vancomycin and aminoglycosides.
Author Response
The manuscript needs serious English revision. It is too laborious to mention all mistakes, but here are some: Abstract Line 12: ..antimicrobials are one of the most frequently used drugs..; line 14: plasmatic; Introduction, line 46: ..improve the achievement Pk/Pd...; Line 51: The most commons Ph changes.. etc.
R: A native-speaker English has reviewed the manuscript. Changes have been made in order to correct grammatical errors, typos and achieve a better understanding.
The authors focus on antibiotics and bacteria only, fungi are not mentioned,, why they should stick to this subject and use this term alone, not antimicrobials, and bacteria in stead of microorganisms.
R: According to this appreciation, we have changed the term antimicrobial for antibiotics
Pharmacokinetics is abreviated to Pk (use PK), then this should be used throughout the text and not introduce the term "pharmacokinetics" several times in the text.
R: We appreciate this suggestion. This change has been included in the manuscript
Table 1 is virtually unreadable. The table should be written vertically to allow space for all columns. Alle abbreviations must be explained either in the legend or in foot notes. The term "f" for "free" (protein non-bound) is only used for T>MIC in the start, and not explained. If this is used, it should also be used for Cmax and AUC in the rest of the table - or do not use it but expalin in the legend that all parameters are "free" concentrations.. fT>100%MIC should be 100% fT>MIC. Ceftazidime is once spelled ceftazidima, and Taz should be written out as tazobactam, since other inhibitors are written out. Explain why T/MIC or T/ 4 x MIC is used ?
R: We agree with this commentary. Table 1 has been redesigned in order to a better understanding. Abbreviations have been included in the footnote.
The manuscript part on prevention of resistance is too simple. The types of resistance for the different drug classes should be explained. It is not clear to me, which type of resistance occurs for beta-lactams (efflux?), while this is more clear for fluoroquinolones. The same goes for resistance towards vancomycin in staphylococci, which is debatable. I thought resistance seldom occurred with gentamicin, while it was common for streptomycin. Also, the importance of high bacterial burden is an issue for the effect of most antibiotics, not just for selection of resistance, but for sheer effect of the antibiotics (persisters and others). In this context it might also be mentioned, that the amount of antibiotics used is also a factor related to selection of resistance, and improving PK/PD should in theory improve the effect and at the same time reduce the duration of therapy, which in itself reduces the exposure on the normal flora.
R: We appreciate these commentaries. We have tried to include these concepts in the manuscript as follows:
Optimization of Pk/Pd parameters to minimize antibiotic resistance has not received sufficient attention in clinical practice. Pharmacokinetic goals are generally focused on maximizing clinical and microbiological outcomes, without considering resistance sup-pression. Nevertheless, given the progressive increase in resistance observed at the usual doses for most antibiotics, clinical data are needed to define thresholds that can minimize the emergence of resistance without compromising patient safety.
Acquired antimicrobial resistance mechanisms can be divided in four main categories: modifying the antibiotic target (PBPs alterations), limiting uptake (decreased numbers of porins), antibiotic inactivation (β-lactamases) and active drug efflux pumps. Gram negative bacteria can use of all four main mechanisms, whereas gram positive bacteria less commonly use limiting the uptake of a drug, and certain types of drug efflux mechanisms are unable. Clinical and in vitro studies have shown that the low antibiotic dose and treatment duration can influence the selection of antibiotic-resistant mutants. [38]. However, the impact of antibiotic dosage or duration in specific mechanisms of resistance expression is uncertain.
There is poor knowledge of the optimal dosing strategies to treat bacterial infections while simultaneously preventing the selection and emergence of resistance. It´s known that a subpopulation of resistant bacteria often exists and can be selected at certain drug concentrations, leading to a regrowth during treatment. Low antibiotic concentrations can select for low-level resistance, which could have a major effect on the emergence of high-level antibiotic resistance [39]. On the other hand, during antibiotic treatment, selection of resistance may take place at several sites. Therefore, in the evaluation of optimal drug concentrations, it is important to focus not only on the infective pathogens and the infectious sites, but also, on the commensal flora (e.g. intestinal tract), in which much higher numbers of bacteria exist and which perhaps is even more important in the selection of resistance
The Pk/Pd index relates antibiotic exposure to the antibiotic susceptibility of an infecting pathogen, in which susceptibility is described as the MIC, thus providing a dosing target. However, in vitro studies simulating current antibiotic dosing highlight the limited ability to suppress the emergence of antibiotic-resistant bacteria [38]. Because the MIC is a measure of susceptibility for most of the bacterial population at a standardized inoculum, alternative measures of susceptibility are needed to determine the risk for devolving antibiotic resistance to provide new Pk/Pd targets for suppressing resistance emergence.
The mutant prevention concentration (MPC), which is defined as the lowest concentration that blocks the emergence of first-step resistant mutants in a large susceptible population [39], has been proposed as a cutoff point to select an antibiotic concentration that prevents antibiotic resistance. Antibiotic concentrations ranging between the MIC and the MPC are known as the mutant selection window (MSW) and promote the growth of resistant bacterial pathogens [40]. Therefore, the antibiotic exposure required to suppress the emergence of resistance should be maintained above the MSW [40]. However, no standardized definitions exist to determine the antibiotic exposures that should be targeted to suppress the emergence of antibiotic resistance. On the other hand, MPC is based on the concept that antibiotic resistance is a function of sequential mutations; therefore, preventing the first mutation will effectively prevent subsequent mutations. The MPC has mostly been studied for antibiotics for which resistance primarily develops by stepwise chromosomal point mutations, especially the fluoroquinolones. The application of MPC determination to other drugs has raised questions regarding the relevance of mutational events for such drugs as the b-lactams and the aminoglycosides and whether MPC measurements can be performed for drugs with other resistance mechanisms (e.g., efflux and b-lactamases) [55–57]However, this rationale may not be applicable in the clinical setting as previous wide-spread use of inferior class agents may already have created a population of first step mutants. Mutations that can lead to resistance may arise in many different combinations, and the correlation between the MIC and the MPC is probably dependent on where the mutation is likely to arise
Table 2: What is the meaning of the "n" after the first MIC in the equation?
R; Table 2 has been corrected, deleting the “n” term
There is very little text on the issue of TDM - only about bedside monitoring. With all the importance of the the PK/PD target it would be logical to discuss the approriateness on more TDM for beta-lactams, while it is standard for vancomycin and aminoglycosides.
R: According to this suggestion, the following paragraph has now been included:
During the last decades, antimicrobial therapeutic drug monitoring (TDM) has evolved from toxicity prevention of drugs with narrow therapeutic index to an essential tool to improve the response to antibacterial treatment and prevent the emergence of resistance. There is now a strong rationale to individualize antimicrobial dosing in critically ill patients with the aid of TDM [9, 10], being positioned as a valuable tool to improve the clinical results of patients with severe infection.
Several studies have shown that the application of TDM leads to improvements in pk/pD achievement, leading to improved clinical outcomes for patients [36,37]. However, the impact of the TDM of new antimicrobials on the clinical evolution of patients and the generation of resistance needs to be evaluated in the coming years, in order to find the best pK/pd values to apply in the different types of infection.
Luxton, T. N., King, N., Wälti, C., Jeuken, L. J. C., & Sandoe, J. A. T. A Systematic Review of the Effect of Therapeutic Drug Monitoring on Patient Health Outcomes during Treatment with Carbapenems. Antibiotics (Basel, Switzerland). 2022, 11(10), 1311.
Cusumano, J. A., Klinker, K. P., Huttner, A., Luther, M. K., Roberts, J. A., & LaPlante, K. L. Towards precision medicine: Therapeutic drug monitoring-guided dosing of vancomycin and β-lactam antibiotics to maximize effectiveness and minimize toxicity. American journal of health-system pharmacy. 2020, 77(14), 1104–1112.

Round 2
Reviewer 1 Report
The authors have revised the manuscript extensively. The language of the text has been corrected also. Having considered the revision, I have minor comments (please see below):
1) The term "pharmacokinetics/pharmacodynamics" is abbreviated in different ways in the text: Pk/Pd and PK/PD. Only one of them should be used, and I recommend the latter one.
2) Page 11, line 534: Please use the Greek letter b here and further where necessary.
3) Page 13, line 660: Did the authors mean AUC/MIC ratio?
4) Page 14: The term Marching learning is still met here which should be corrected too.
5) Table 1 is absolutely incomprehensible when the corrections are shown. Please provide it again.
Author Response
1) The term "pharmacokinetics/pharmacodynamics" is abbreviated in different ways in the text: Pk/Pd and PK/PD. Only one of them should be used, and I recommend the latter one.
R: This term has been homogenized in the manuscript to Pk/Pd
2) Page 11, line 534: Please use the Greek letter b here and further where necessary.
R: We appreciate your suggestion. We have included this correction in the manuscript
3) Page 13, line 660: Did the authors mean AUC/MIC ratio?
R: We appreciate your correction. It has been changed to AUC/MIC
4) Page 14: The term Marching learning is still met here which should be corrected too.
R: ”The term “Marching” has been corrected.
5) Table 1 is absolutely incomprehensible when the corrections are shown. Please provide it again.
R: Table 1 has been submitted again
Reviewer 2 Report
The manuscript was revised according to the reviewers’ comments.
Author Response
The manuscript was revised according to the reviewers’ comments.
R: We appreciate your suggestions.
Reviewer 3 Report
The authors have responded adequately to most comments; there are still some places, where anitmicrobial" has not been corrected to "antibiotic", and "Marching" should be corrected to "Machine". Otherwise ok.
Author Response
The authors have responded adequately to most comments; there are still some places, where anitmicrobial" has not been corrected to "antibiotic", and "Marching" should be corrected to "Machine". Otherwise ok.
R: We apologize for the use of antimicrobials in the last version. We have reviewed the manuscript again and changed to “antibiotic”. The term “Marching” has also been corrected.